# Economic Uses of Salt-Tolerant Plants

**DOI:** 10.3390/plants12142669

**Published:** 2023-07-17

**Authors:** Pedro Garcia-Caparros, Mohammed J. Al-Azzawi, Timothy J. Flowers

**Affiliations:** 1Department of Superior School Engineering, University of Almería, Ctra. Sacramento s/n, La Cañada de San Urbano, 04120 Almería, Spain; pgc993@ual.es; 2School of Life Sciences, University of Sussex, Falmer, Brighton BN1 9QG, UK; ma2442@sussex.ac.uk

**Keywords:** plant salt tolerance, halophytes, economic uses, eHALOPH

## Abstract

Climate change is likely to affect the ability of world agricultural systems to provide food, fibre, and fuel for the growing world population, especially since the area of salinised land will increase. However, as few species of plants (less than 1% of all plant species) can tolerate saline soils, we believe it is important to evaluate their potential as crops for salinised soils. We have analysed the economic and potential economic uses of plants that are listed in the database eHALOPH, including the most tolerant species, halophytes. For nine main categories of economic value, we found a total of 1365 uses amongst all species listed in eHALOPH as of July 2022; this number reduced to 918 amongst halophytes. We did not find any obvious differences in rankings between the more tolerant halophytes and the whole group of salt-tolerant plants, where the order of use was medical, followed by forage, traditional medicine, food and drink, fuel, fuelwood, and bioenergy. While many species are potentially important as crops, the effects of salt concentration on their uses are much less well documented. Increasing salt concentration can increase, decrease, or have no effect on the concentration of antioxidants found in different species, but there is little evidence on the effect of salinity on potential yield (the product of concentration and biomass). The effect of salinity on forage quality again varies with species, often being reduced, but the overall consequences for livestock production have rarely been evaluated. Salt-tolerant plants have potential uses in the bioremediation of degraded land (including revegetation, phytoremediation, and extraction of NaCl) as well as sources of biofuels, although any use of saline water for the sustainable irrigation of salt-tolerant crops must be viewed with extreme caution.

## 1. Introduction

### 1.1. Population and Food

Recent predicted changes to the climate and their consequent effects on food production have generated thousands of publications over the last five years (3125 papers found in searching the research topic of ‘review and (world or global) and (food and production)’ in the Web of Science core collection in mid-October 2022) and focused minds on feeding the human population of the world in the future. Twelve years ago, FAO had already concluded that “The world has the resources and technology to eradicate hunger and ensure long-term food security for all, in spite of many challenges and risks. It needs to mobilize political will and build the necessary institutions to ensure that key decisions on investment and policies to eradicate hunger are taken and implemented effectively” (https://www.fao.org/fileadmin/templates/wsfs/docs/expert_paper/How_to_Feed_the_World_in_2050.pdf (accessed on 14 June 2023)). More recently, van Dijk et al. [1] conducted a meta-analysis of 57 studies published between 2000 and 2018 that enabled them to predict increases in global food demand and the global population at risk of hunger. Global food demand was forecast to increase by between 35% and 56% by 2050 compared with 2010. Achieving such increases may require new food ‘frontiers’, “innovations in food production technologies and techniques” [2], especially following the worrying words in the executive summary of the FAO document on feeding the world (*loc. cit*.) that the rate of growth of yield of the main cereals had fallen to about 1.5% per year in 2000. In our view climate change is likely to have a negative impact on food production in particular areas of the world through its effects on salinisation, since high concentrations of salts in the soil generally decrease plant growth (see below).

### 1.2. Salinisation

Many estimates have been made over the years of the extent of salt-affected land and its economic consequences. Past estimates of the area affected ranged from 340 to 950 million ha [3]. The recently launched website of the Global Framework on Water Scarcity in Agriculture (WASAG) shows that over 424 million hectares (MHa) of topsoil (0–30 cm) and 833 million hectares of subsoil (30–100 cm) are currently salt-affected (https://www.fao.org/global-soil-partnership/gsasmap/en accessed on 14 June 2023). Data in the map is supported by an analysis carried out by Hassani et al. [4], who estimated 590 MHa of land suffered with an electrical conductivity of a saturated extract of the soil (ECe) ≥ 4 dS·m^−1^ in ¾ of the years between 1980 and 2018. These are large areas of land, that include what were highly productive irrigated soils [5] and are likely to be made even larger by the consequences aridity, with its need for irrigation for agricultural production, and rising sea levels in coastal regions. 

The oceans of the world cover about 71% of the world’s surface with some 356,000 km of coastline (https://www.citypopulation.de/en/world/bymap/coastlines/ accessed on 14 June 2023). These coastlines are the zones in contact with the salt solution that makes up the oceans of the world and so their immediate hinterlands are the first in line for any inundation that might be caused by tsunami [6,7], storm surges [8] or extremes in tidal height, see [9]. Such events have occurred over the centuries but might be expected to occur with greater frequency as a consequence of global warming and the rise in sea level; the reasons are discussed in detail by [10].

As well as direct salinisation of land by seawater, climate change is predicted to increase aridity in some parts of the world [11]. In arid regions, agriculture relies heavily on irrigation [5], as plants require water to be transpired in order to grow. However, since, over time, irrigation commonly results in the build-up of salts in the soil profile, e.g., [12,13], yields tend to decline and land goes out of production as the majority of our plants and crops are sensitive to salt.

### 1.3. Salinisation, Plant Growth, and Crop Yields

More than sixty years ago, it was clear that most of our crops are sensitive to salt [14] and although data on differential tolerance was scarce [15], variations in tolerance between varieties had been described, e.g., [16,17,18]. By the mid-1970s, comprehensive lists of crop tolerance were published [19]; see [20] for a more comprehensive history of the approaches to evaluate salt tolerance in crops. By the 1980s, approaches to breeding for salt tolerance were being explored [21,22] as well as the use of halophytes in forage production [23]. A comprehensive list of differences in salt tolerance of crop species was published by Maas [24]. Tolerance was characterised by a threshold up to which no loss of yield occurred and a subsequent rate of yield loss per unit increase in salinity estimated as the conductivity of a saturated extract of the soil solution—although many of the rankings were determined for a limited range of varieties and under specific agronomic conditions; see [25,26] for potential improvements to the methodology.

### 1.4. Ways to Deal with the Salinity Problem

At around the time crop sensitivity to salt was being documented, the use of saline water in agriculture was promulgated by Hugo and Elisabeth Boyko [27,28,29]. They claimed to be able to irrigate crops with salt water provided the substrate was sand or gravel. However, this use of seawater to raise conventional crops has not, as far as we are aware, been a success and reported results have not allowed evaluation of productivity, the importance of fog or rain or of irrigation frequency [30]. Halophytes, however, can grow in saline soils and by 1969, their potential use as forage was shown by the pioneer Clive Malcolm, e.g., [31]. Mudie [30,32] was able to demonstrate that salt tolerant plants could be raised with seawater. Subsequently, Aronson built on her work and in a seminal publication [33] compiled a list of over 1560 species “complied for anyone growing or planning to grow halophytes” based on the primary criterion of a “known or presumed tolerance to electrical conductivity measuring (or estimated to be) at least 7.8 dS m^−1^, during significant periods of the plant’s entire life” [33]. The potential of halophytes as new vegetable, forage, and oilseed crops as well as in managing saline agricultural wastewater was explored in a review published towards the end of the last century by Ed Glenn, another pioneer in the field, with his colleagues [34]. There have been very many other reviews of the use of various aspects of using saline water and salt-tolerant plants in agriculture, horticulture, and the remediation of degraded lands; here we provide examples (and we emphasize these are just examples) from the last ten years include; Cassaniti et al. [35], Ventura and Sagi [36], Rozema and Schat [37], Hasanuzzaman et al. [38], Nizar et al. [39], Loconsole et al. [40], Li et al. [41], Oliveira et al. [42], Holguin-Pena et al. [43], Spradlin and Saha [44], and García-Caparros et al. [45]. 

## 2. eHALOPH and the Current Review

In the current review, we have looked at the economic uses, established and potential, of salt-tolerant plants. These include those that tolerate a salt concentration equivalent to about 80 mM NaCl and those tolerating at least 200 mM NaCl (halophytes). To do this, we have used the data collected in Aronson’s HALOPH, which have been incorporated into an electronic database that is available at https://ehaloph.uc.pt/ accessed on 14 June 2023, see also [46]. The data has been reviewed and extended (by Flowers and Al-Azzawi https://ehaloph.uc.pt/ accessed on 14 June 2023), so that there are now about 1200 salt-tolerant species in the database in 93 families with 650 halophytes in 46 families. If families are ranked by the number of species, then the Amaranthaceae, Poaceae, Fabaceae, and Plumbaginaceae dominate both groups. The Zosteraceae, Cymodocaeae, and Acanthaceae are ranked higher amongst halophytes than in all salt-tolerant plants (Table 1).

We have used the data in eHALOPH to construct tables of use by family, genus, and species of all the species listed in the database. We searched the Web of Science (all databases) for “*Genus species*” and (economic or use), in ‘Topic’ or ’Abstract’ or ‘Title’ in order to find a consensus of uses for the species and add one or two supporting references. The economic uses were classified according to a modified version of The Survey of Economic Plants for Arid and Semi-Arid Lands (SEPASAL; http://sftp.kew.org/pub/data-repositories/sepasal/ accessed on 1 June 2022) used by Aronson [33]. The main categories are shown in Table 2 and the full list used in eHALOPH is in Appendix A.


Main economic uses


For the nine main categories (Table 2), we found a total of 1365 uses amongst all species listed in eHALOPH as of July 2022; this number reduced to 918 amongst halophytes (Table 3). For both groups of plants, most uses (close to 30%) were medical followed by use as forage (20%). 

There were no obvious differences between the more tolerant halophytes and the whole group of salt-tolerant plants, where the order of use was medical, followed by forage, traditional medicine, food and drink, fuel, fuelwood, and bioenergy (Table 3); 12 crops (classified as crops according to a list published by FAO at https://www.fao.org/fileadmin/templates/ess/documents/world_census_of_agriculture/appendix4_r7.pdf accessed on 14 June 2023) were found to be salt-tolerant. Of the 404 species with medical uses, 201 were used as traditional medicines (Category 7100). Amongst the other categories there were 91 species with uses as fuel, fuelwood, or bioenergy (2100 + 2110 + 2120 + 8300). Clearly, the physiology of salt tolerance in plants has not limited their use by humans. In the following analyses, we have used data from all the species listed in eHALOPH, rather than restricting our analysis to halophytic (species at least 200 mM NaCl), to illustrate how economic use and salt tolerance can be complementary traits. We have not sought to show every economic use of these salt-tolerant plants, but to illustrate their potential value in a salinizing world.


Medical (Traditional medicine)


Salt-tolerant plants have long been known for their uses as traditional medicines (see [48] for an extensive list) and examples are provided by Buhmann and Papenbrock [49] in their wide-ranging review of compounds from halophytes with potential pharmacological value. Their review highlighted the potential of halophytes for the production of pharmaceuticals including compounds with antioxidative and antimicrobial activity, with cautionary notes on the importance of extraction methods, plant identification, and the identification of bioactive chemical constituents. Buhmann and Papenbrock [49] give examples of increases in phenols and flavonoids with increasing external salinity, a trait that could be exploited were halophytes to be grown for their chemical content. 

We have found 90% of (404) medical uses occurred in thirty-six families; 50% of medical uses are accounted for by nine families (Table 4). Of these, a high proportion of salt-tolerant species in the Apiaceae, Rhizophoraceae, and Malvaceae families are used as traditional medicines. These medicinal uses presumably relate to antibiotic properties and/or the content of specific chemicals within the plants. For example, Rodrigues et al. [50] have reviewed the antiparasitic properties of halophytes and highlight the anthelmintic properties of Dysphania ambrosioides.

In order to evaluate if external salinity influences the medicinal properties of salt-tolerant plants, we have looked at 102 papers on 65 species from 27 families within the whole range tolerance of species included in eHALOPH to determine if growth under saline conditions alters the content of antibiotic compounds (Category 7160.0 in Table 1), but could not find evidence that this had been investigated. Our conclusion mirrors that of Selmar [51] who looked at the concentrations of pharmaceuticals produced by plants under drought and salt stress. He noted that the effects of the stress on growth is likely to negate any increase in quantity of the compound that might be produced. For salt-tolerant plants, loss of biomass in the presence of salinity, may not be so much of an issue, as growth can even be promoted by salt [47]. Further investigations of the effects of salinity on antibiotic content of salt-tolerant plants is clearly an area worth further investigation.


Antioxidants


The medicinal properties of plants might not only reflect the presence of antibiotics but also of antioxidants, see [49] mentioned above. Consequently, we have tabulated the number of families of salt-tolerant plants reported to contain potential antioxidants: flavonoids, together with phenols and polyphenols are reported in over 100 species and tannins in about 20 species (Table 5). 

Although flavonoids have been reported from 116 species of salt-tolerant plant, we found just 20 species where the effects of external salinity on their concentration was recorded (in 12 papers). It is clear from the data in Table 6 that salt can increase (12 species), or have no effect on (6 species) or decrease (2 species) the concentration of flavonoids.

For phenols and polyphenols, we found 75 papers reporting the concentrations of phenols or polyphenols, of which 13 reported the effects of salinity on the concentration in 22 species (Table 7). Salinity increased the phenol concentration in thirteen species, had no or little effect in four, and decreased the concentration in five species.

Although tannins have been measured in at least 21 species of salt-tolerant plants Table 5 [72], we did not find any examples of reports of the effects of salinity on their concentration. 

In most cases where concentrations of antioxidants are reported, they are on a dry weight basis, but where a fresh weight basis was used care needs to be taken that the results were not influenced by a changing water content. As far as obtaining an economic yield of flavonoids, polyphenols, or tannins is concerned, it is important to understand that while concentrations can be quite variable between species e.g., [56], yields will depend on the weight of the plant that can be harvested and on the effects of salt concentration on that biomass. Rarely has the combination of salt and successive harvests been investigated, but see [61]: it can mean that maximal yields are achieved in fresh water, e.g., *Polygonum maritimum* [61]. There appears to be a weak correlation between salinity and the concentration of phenols and flavonoids in plants occupying habitats ranging from salt marsh to dunes, a semiarid calcareous area, and a gypsiferous area [73], interpreted as showing halophytes are not under oxidative stress in their natural habitats. Interestingly, the concentrations of polyphenols were significantly higher in mangrove trees than in herbaceous halophytes growing in neighbouring sites [74].


Forage


Salt tolerant plants have a long history of use as forage or fodder particularly as reserve feed, see [75,76,77] and have considerable potential in the restoration of degraded land [78]. For forage, various measures are used to assess quality, these include acid detergent fibre (ADF), acid detergent lignin (ADL), ash content, crude fibre (CF), crude protein (CP), in vitro organic matter digestibility (IVOMD), metabolizable energy (ME), organic matter digestibility (MD), and neutral detergent fibre (NDF), see, for example, [76,79,80], with a range of methods being available to estimate in vivo digestibility, see for example [81]. 

After medical use, forage has the highest economic use found for salt-tolerant plants (Table 8), but unlike medical use, just three families dominate use as forage and only 31 of the 93 families have any recorded use as forage.

In an extensive review based mainly on research conducted in the Near East, El Shaer et al. [76] concluded that halophytes have potential as feedstuff for sheep, goats, and camels. Halophytes have also been shown to have potential for the production of forage in Iran [82] and California, e.g., [83,84], as in other arid regions [80,85], although quality can be low, e.g., [86]. 

Although there are many papers suggesting the value of halophytes as forage, there is less information on the effect of salinity on the quality of that forage. We found data on 18 species, which mostly suggested that forage quality was maintained as salinity increased (Table 9). However, in assessing the feed quality of any particular halophyte, care should be taken as a high (perhaps 50%) of the dry matter can be salt, requiring the animals to drink more water and the organic dry matter can have low digestibility and contain toxic elements such as oxalate [77,87]. Advantages in using halophytes may come from the nitrogenous compounds used in osmotic adjustment [47] and minerals and metabolites associated with antioxidant activity, see for example., [87,88]. Consequently, assessing the value simply in terms of dry matter production is inadvisable. 


Bioremediation


The data in eHALOPH show 269 species are used for what is termed ‘bioremediation’ (or have potential use in bioremediation) with 81% being accounted for in fourteen families (Table 10): three families account for 52% of uses or potential uses. We have analysed the potential uses of the Amaranthaceae, for which we found 63 records with most being for revegetation (30 species): other uses were extraction of NaCl (20), phytoremediation, (18) and extraction of heavy metals (14).


Biofuels


The use of halophytes as sources of biofuels is important in the context of the competition for land for the production of food and fuel. The continuous rise in atmospheric CO_2_ is a driver for a reduction in the use of fossil fuels and their eventual replacement by fuels generated from plants [101,102], thus reducing the introduction of stored CO_2_ into the atmosphere and recycling that which is already present. To date, feedstocks for biofuels have been edible crops such as sugarcane, wheat, maize, soybean, and rapeseed, but the need to feed the human population of the world has led to the development of second-generation feedstocks, Miscanthus and Jatropha, but which still require the use of land that could grow food. Consequently, interest has grown in the use of third-generation feedstocks—plants that can grow on saline land and/or be irrigated with salt water. Salt-tolerant plants offer a new opportunity for biofuel production where competition for land use for food is minimized, see [103]. The data in eHALOPH suggest that just 10 families offer more than 80% of the 97 species with economic uses as fuel, fuelwood, charcoal, or bioenergy (Table 11).

## 3. Conclusions

The data in eHALOPH has been primarily produced from that in Aronson [33], supplemented with that listed by [104,105,106], although only those species where there is clear published evidence of tolerance of the equivalent of about 80 mM NaCl are included (there are likely to be many other species that are salt tolerant, but without published verification). Within the data in eHALOPH, it is possible to separate halophytes as those tolerating 200 mM NaCl or more, a soil salinity of at least 20 dS m^−1^ (in the saturation extract), sea water or its equivalent salt concentration of 35.5 g L^−1^. For all those species meeting the criterion of salt tolerance, we made a specific search for economic uses by searching all database in the Web of Science for “*Genus species*” and (economic or use) for all the species in the database. This was completed in July 2022.

The importance of salinity for food production has been known for thousands of years [107] as has the consequent need to understand and increase the salt tolerance of our major crops, see [20]. However, there is likely a new urgency to exploit salt-tolerant plants given changes in climate and human population. Almost thirty years ago, Flowers and Yeo [3] posed the “whether, at least in some cases, effort would not be better spent on domestication of halophytes rather than on improving resistance of conventional crops”. It is in this context that the various potential uses of salt tolerance should be viewed. Flowers and Yeo [3] advocated the use of halophytes in seriously salt affected soils, concluding “As O’Leary [108] points out, it is the improvement of agronomic characteristics that has the history of success; the enhancement of salt resistance does not”. While there have been improvements in the salt resistance of major crops such as rice [109,110] and wheat [111,112], it remains the case that where salinity is entrenched, salt tolerant plants offer the most likely means to revegetate land and obtain useful yields of plant material whether for food, fibre or fuel, see [113,114] for reviews of the uses of salt-tolerant plants in agriculture. Salt-tolerant plants can also be a source of plant growth promoting rhizobacteria that may aid the growth of less salt-tolerant plants under saline conditions [115]. We have shown that there are species that can be grown in what for our current crops would be toxic conditions.

While salinity has a detrimental effect on the yield of most of our conventional crops, it is clear that growth rates of halophytes in saline conditions that match those of less salt-tolerant species, see [34]; this is true of halophytic trees [116]. As a consequence, the use of salt-tolerant forages (Table 8) has many supporters, but there is too little information to be certain about the effects of salinity on the quality of forage. Similarly, there is a paucity of information on the effects of salinity on antioxidants, such as flavonoids, phenols and tannins. If salt-tolerant plants are to be grown for their chemical contents, then it will be important to evaluate the effects of external salt concentration on the yield—a combination of biomass and concentration. The concentration can increase under saline conditions (e.g., flavonoids, Table 6), but yield depends on the response of biomass harvested. 

While it may be possible to utilise halophytes for forage and even pharmaceuticals on naturally or soil salinised by human activity, whether they can be produced with saline irrigation is a different question [117]. Saline water dramatically affects soil structure, so the sustainable use of saltwater for irrigation is only likely on sandy soils with deep drainage and even here the long-term consequences of an increase of saline groundwaters must be considered. Halophytes are likely to have a place in both agriculture and horticulture, but in carefully managed situations.

## Figures and Tables

**Table 1 plants-12-02669-t001:** Ranking of families of salt-tolerant plants (those tolerating an electrical conductivity of a saturated soil past, ECe, of at least 7.8 dS m^−1^) and halophytes (defined as those plants with an ability to grow in at least 200 mM NaCl [47]; we also approximated this to a conductivity of 20 dS m^−1^) in the database eHALOPH as of July 2022. The 20 families listed accounted for 81% of both all species of salt-tolerant plants and of halophytes.

	All Salt-Tolerant Plants	Rank	Halophytes
1	Amaranthaceae	1	Amaranthaceae
2	Poaceae	2	Poaceae
3	Fabaceae	3	Fabaceae
4	Plumbaginaceae	4	Plumbaginaceae
5	Asteraceae	5	Zosteraceae
6	Cyperaceae	6	Asteraceae
7	Tamaricaceae	7	Hydrocharitaceae
8	Hydrocharitaceae	8	Rhizophoraceae
9	Myrtaceae	9	Cymodoceaceae
10	Aizoaceae	10	Myrtaceae
11	Zosteraceae	11	Acanthaceae
12	Brassicaceae	12	Aizoaceae
13	Rhizophoraceae	13	Cyperaceae
14	Solanaceae	14	Brassicaceae
15	Cymodoceaceae	15	Tamaricaceae
16	Malvaceae	16	Malvaceae
17	Acanthaceae	17	Posidoniaceae
18	Apiaceae	18	Casuarinaceae
19	Arecaceae	19	Lythraceae
20	Zygophyllaceae	20	Solanaceae

**Table 2 plants-12-02669-t002:** The main categories of economic uses used in eHALOPH and based on The Survey of Economic Plants for Arid and Semi-Arid Lands (SEPASAL; http://sftp.kew.org/pub/data-repositories/sepasal/ accessed 1 June 2022) used by Aronson [33].

Code	Use	Sub Code	Use
000.0	FOOD AND DRINK	0100.0	Vegetables and fruit
		0200.0	Beverages
		0300.0	Cooking fats and oils
		0400.0	Miscellaneous food and drinks
		0500.0	Breeding stock
	CROP LISTED BY FAO	0001.0	
1000.0	DOMESTIC PRODUCTS	1200.0	Soaps
		1300.0	Cosmetics
		1400.0	Dental
		1700.0	Roofing thatching and green roofs
2000.0	TIMBER	2100.0	Fuel
		2200.0	Sawn timber
		2400.0	Construction timber
		2500.0	Carpentry
3000.0	FORAGE	3100.0	Grazing
		3200.0	Browse
		3300.0	Fodder
4000.0	LAND USE	4500.0	Soil stabilization
		4600.0	Soil improvement
		4700.0	Salt tolerance
		4800.0	Ornamental
5000.0	FIBERS	5100.0	Cordage
		5200.0	Textiles
6000.0	TOXINS	6000.0	
7000.0	MEDICAL	7100.0	General including traditional medicine
		7160.0	Antibiotics
8000.0	CHEMICALS	8100.0	Carbohydrates
		8200.0	Lipids, Essential oils
		8300.0	Bioenergy/Biofuel

**Table 3 plants-12-02669-t003:** The economic uses of salt-tolerant plants (all species in eHALOPH) and halophytes as defined by Flowers and Colmer [47], see Table 1. The percentage values are of total uses. The numbers in columns ‘All’ and ‘Halophytes’ are for all publications in the main and subcategories of the SEPASAL divisions—see Table 1 and Appendix A.

SEPASAL Code	SEPASAL Descriptor	All	%	Halo-Phytes	%
0	FOOD AND DRINK; Crops	152	11.1	107	11.7
1000	DOMESTIC PRODUCTS	39	2.9	27	2.9
2000	TIMBER	80	5.9	56	6.1
3000	FORAGE	275	20.1	189	20.6
4000	LAND USE	206	15.1	149	16.2
5000	FIBERS	27	2.0	14	1.5
6000	TOXINS	42	3.1	25	2.7
7000	MEDICAL	404	29.6	258	28.1
8000	CHEMICALS	140	10.3	93	10.1
	Totals	1365	100	918	100

**Table 4 plants-12-02669-t004:** Families of salt-tolerant plants with the highest number of species with reported medical use as recorded in eHALOPH in July 2022.

Family	Number	%	Sum %	% All Species in Family
Fabaceae	50	12		54
Amaranthaceae	42	10	23	13
Asteraceae	24	6	29	47
Plumbaginaceae	21	5	34	38
Poaceae	20	5	39	13
Rhizophoraceae	16	4	43	84
Solanaceae	12	3	46	67
Apiaceae	11	3	49	85
Malvaceae	11	3	51	73

**Table 5 plants-12-02669-t005:** The number of families of salt-tolerant plants reported to contain flavonoids, phenols, polyphenols, and tannins.

Metabolite	Flavonoids		Phenols and Polyphenols		Tannins	
Number of families with metabolite	35 (116 spp)		36 (102 spp)		11 (21 spp)	
Families accounting for at least 70% of occurrences and % of total reported for the metabolite	Amaranthaceae	18	Amaranthaceae	22	Amaranthaceae Rhizophoraceae Acanthaceae Aizoaceae Fabaceae	27 19 12 8 8
	Plumbaginaceae	10	Fabaceae	7		
	Fabaceae	9	Plumbaginaceae	7		
	Asteraceae	9	Asteraceae	6		
	Hydrocharitaceae	3	Rhizophoraceae	6		
	Poaceae	3	Brassicaceae	4		
	Rhizophoraceae	3	Poaceae	4		
	Aizoaceae	3	Acanthaceae	3		
	Apiaceae	3	Aizoaceae	3		
	Apocynaceae	3	Arecaceae	3		
	Brassicaceae	3	Cymodoceaceae	3		
	Cyperaceae	3	Malvaceae	3		

**Table 6 plants-12-02669-t006:** The effect of salinity on the concentration of flavonoids in salt-tolerant plants. DW, dry weight; FW, fresh weight.

Family	Genus Species	Effect of Salt on Concentration of Flavonoids	References
Aizoaceae	*Sesuvium portulacastrum*	200 and 500 mM NaCl increased some and decreased others of a large range of flavonoids (DW basis)	[52]
Amaranthaceae	*Halocnemum strobilaceum*	No significant effect with increasing soil K_2_SO_4_ to 135 kg ha^−1^ soil (presumed DW basis)	[53]
Amaranthaceae	*Salicornia brachiata*	200 and 500 mM NaCl increased a large range of flavonoids (DW basis)	[52]
Amaranthaceae	*Suaeda fruticosa*	No effect of 500 mM NaCl on kaempferol at 400 ppm CO_2._ A 10 -fold decrease in kaempferol in 500 mM NaCl at 900 ppm CO_2_ (presumed DW basis)	[54]
Amaranthaceae	*Suaeda maritima*	200 and 500 mM NaCl increased some and decreased others of a large range of flavonoids (DW basis)	[52]
Amaranthaceae	*Suaeda monoica*	A 10-fold increase in kaempferol in 500 mM NaCl at 900 ppm CO_2_ (presumed DW basis)	[54]
Apocynaceae	*Apocynum venetum*	Over the range 50 to 400 mM NaCl, kaempferol and quercetin increased to maximal values at 100 mM NaCl (DW basis)	[55]
Asteraceae	*Tripolium pannonicum*	No effect of 15 or 50 PSU over 5 weeks (FW basis)	[56]
Brassicaceae	*Lepidium latifolium*	Increased over 24 h exposure to 30 PSU (FW basis)	[56]
Fabaceae	*Sulla carnosa*	No effect of 100 mM NaCl (DW basis)	[57]
Lamiaceae	*Dracocephalum kotschyi*	Increased over the range 25–75 mM NaCl then declined at 100 mM (DW basis)	[58]
Nitrariaceae	*Nitraria schoberi*	Increased with increasing soil K_2_SO_4_ to 135 kg ha^−1^ soil (presumed DW basis)	[53]
Plantaginaceae	*Plantago coronopus*	Increased (by 74%) over 5 weeks at 15 PSU (FW basis)	[56]
Plumbaginaceae	*Limonium bicolor*	No effect as NaCl increased to 200 mM; decreased at 300 mM NaCl (DW basis)	[59]
Plumbaginaceae	*Limonium delicatulum*	Small increase to 200 mM NaCl over the range 50 to 500 mM; then a decline (DW basis)	[60]
Polygonaceae	*Polygonum maritimum*	Declined as NaCl increased to 300 mM (DW basis)	[61]
Rhamnaceae	*Colubrina asiatica*	Increased from 0 through 100, 200 and 300 mM NaCl (DW basis)	[62]
Rhizophoraceae	*Bruguiera cylindrica*	No effect of 15 or 50 PSU over 7 weeks (FW basis)	[56]
Rhizophoraceae	*Kandelia candel*	Increased from 0, through 200 and 500 mM NaCl (FW basis)	[63]

**Table 7 plants-12-02669-t007:** The effect of salinity on the concentration of phenols and polyphenols in salt-tolerant plants. DW dry weight; FW fresh weight.

Family	Genus Species	Effect of Salt on Concentration of Phenols and Polyphenols	References
Aizoaceae	*Mesembryanthemum crystallinum*	No effect of seawater concentrations with EC values of 2, 4, 8, 12, 16, 20, and 35 dS m^−1^	[64]
Aizoaceae	*Mesembryanthemum edule*	Effect of 300 and 600 mM NaCl checked on range of phenols and polyphenols in two provenances. Polyphenols declined in both (DW basis)	[65]
Aizoaceae	*Sesuvium portulacastrum*	200 mM NaCl had no effect on total polyphenols in leaves, but concentrations (DW basis) were reduced in stems and roots	[66]
Amaranthaceae	*Atriplex halimus*	Decreased over 5 weeks at 15 PSU (FW basis)	[56]
Amaranthaceae	*Atriplex portulacoides*	Decreased over 5 weeks at 15 PSU (FW basis)	[56]
Amaranthaceae	*Atriplex prostrata*	Total phenols (DW basis) increased in leaves from 10 through 100 to 200 and 300 mM NaCl	[67]
Amaranthaceae	*Salicornia dolichostachya*	Decreased over 5 weeks at 15 PSU (FW basis)	[56]
Amaranthaceae	*Salicornia neei*	No effect of NaCl concentrations up to 769 mM on phenolics of two genotypes	[68]
Apiaceae	*Crithmum maritimum*	Small effects of 50 or 100 mM NaCl on 4 genotypes: total polyphenols either increased, decreased of were unaffected	[69]
Asteraceae	*Tripolium pannonicum*	No effect of 15 PSU over 5 weeks Increased to 22.5 PSU; no further increase at 30 PSU (FW basis)	[56]
Brassicaceae	*Lepidium latifolium*	Increased over 24 h exposure to 30 PSU. Decreased over 5 weeks at 15 PSU (FW basis)	[56]
Fabaceae	*Acacia stenophylla*	Phenols increased by 156% at seawater salinity increased from 0.6 to 16.7 dS m^−1^ (FW basis)	[70]
Fabaceae	*Sulla carnosa*	Total polyphenols increased in 100 mM NaCl (DW basis)	[57]
Lamiaceae	*Dracocephalum kotschyi*	Increased over the range 25–75 mM NaCl then declined at 100 mM (DW basis)	[58]
Lamiaceae	*Vitex trifolia*	Elevating salinity to 5000 ppm increased concentration of total phenols compared to 260 ppm (DW basis)	[71]
Plantaginaceae	*Plantago coronopus*	Total phenols (DW basis) increased in leaves from 10 through 100 to 200 mM NaCl then no change at 300 mM	[67]
Plantaginaceae	*Plantago coronopus*	Increased over 5 weeks at 15 PSU (FW basis)	[56]
Plumbaginaceae	*Frankenia pulverulenta*	Total phenols (DW basis) increased in leaves from 10 through 100 to 200 mM NaCl then no change at 300 mM.	[67]
Plumbaginaceae	*Limonium bicolor*	Phenols decreased (79%) as NaCl increased to 300 mM NaCl (DW basis)	[59]
Rhizophoraceae	*Bruguiera cylindrica*	Increased from 15 to 50 PSU over 7 weeks (FW basis)	[56]
Rhizophoraceae	*Kandelia candel*	Increased from 0, through 200 and 500 mM NaCl (FW basis)	[63]

**Table 8 plants-12-02669-t008:** Families of salt-tolerant plants with the highest number of species with reported use as forage as recorded in eHALOPH in July 2022.

SEPASAL Codes	Forage 3000	Grazing 3100	Browse 3200	Fodder 3300	All	%
Amaranthaceae	36	22	6	40	104	38
Poaceae	38	26	1	19	84	68
Fabaceae	24	6	1	12	43	84
Cyperaceae	3	1	0	0	4	85
All families	112	64	13	86	275	100

**Table 9 plants-12-02669-t009:** Examples of halophytes where measures of the effects of salinity on quality have been evaluated. Data followed the review of 192 publications on 108 species.

Species	Effect of Increasing Salinity on Quality	References
*Atriplex amnicola*	Some quality measures reduced; was genotypic variation	[79]
*Atriplex gardneri*	Forage quality much reduced	[89]
*Atriplex leucoclada*	Higher quality at lower salinity site and dropped with age	[90]
*Atriplex nummularia*	Little effect of salinity on yield or nutrients	[91]
*Bassia prostrata*	Forage quality reduced	[89]
*Cenchrus ciliaris*	Tolerant genotypes identified with adequate nutritive value	[92]
	Carbohydrate, protein, and micronutrients increased per unit of dry weight	[93]
*Diplachne fusca (Leptochloa fusca)*	Little effect on crude protein, fibre, fat, and soluble carbohydrate on dry weight basis	[94]
*Lotus tenuis*	Yield decreased but carbohydrates more digestible	[95]
*Melilotus albus*	Digestible dry matter declined but fibre maintained at high salinity	[96]
*Melilotus indicus*	No digestible dry matter beyond 80 mM NaCl but fibre maintained	[96]
*Melilotus siculus*	Digestible dry matter retained but fibre decreased at high salinity	[96]
*Pennisetum clandestinum*	The percentage of crude protein and crude fibre preserved	[97]
*Salicornia europaea*	Seed oil content increased	[98]
*Salsola tragus*	Total nitrogen increased but fibre decreased on a dry weight basis	[99]
*Sporobolus pumilus (Spartina patens)*	Little effect on crude protein, fibre, fat, and soluble carbohydrate on dry weight basis	[94]
*Sporobolus virginicus*	Little effect on crude protein, fibre, fat, and soluble carbohydrate on dry weight basis	[94]
*Suaeda vermiculata*	Higher quality at lower salinity site and dropped with age	[90]
*Tripolium pannonicum*	Little effect on total nitrogen or micronutrients	[100]

**Table 10 plants-12-02669-t010:** Families of salt-tolerant plants used for or with potential uses in bioremediation. The figures are the number of species for which there is at least one record in eHALOPH of use for bioremediation and the percentage show the proportion of the total 269 species.

Families	No.	%
Amaranthaceae	63	23
Poaceae	53	20
Fabaceae	24	9
Cyperaceae	12	4
Asteraceae	10	4
Tamaricaceae	9	3
Rhizophoraceae	8	3
Myrtaceae	7	3
Aizoaceae	6	2
Juncaceae	6	2
Zosteraceae	6	2
Brassicaceae	5	2
Casuarinaceae	4	1
Hydrocharitaceae	4	1

**Table 11 plants-12-02669-t011:** Families of salt-tolerant plants used to provide fuel, fuelwood, charcoal, or bioenergy. The figures are the number of species for which there is at least one record in eHALOPH of use for one of the uses and the percentage show the proportion of the total 97 species.

Families	Number	Percentage
Fabaceae	14	15
Amaranthaceae	12	13
Poaceae	10	11
Rhizophoraceae	9	10
Tamaricaceae	9	10
Arecaceae	7	7
Casuarinaceae	5	5
Myrtaceae	5	5
Combretaceae	4	4
Malvaceae	4	4

## Data Availability

The data used to construct the tables in this review are available at https://ehaloph.uc.pt/. and were accessed between 1 July 2022 and 22 June 2023.

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
