# Peer review of "Economic Uses of Salt-Tolerant Plants"

_plants, 2023, doi:10.3390/plants12142669_

Round 1

Reviewer 1 Report

The review article titled “Economic uses of salt-tolerant plants” by Garcia-Caparros et al. is an overview based mainly on the database eHALOPH to describe the salinity problem and economic uses of salt-tolerant plants (mainly halophytes). Although the review topic is covered in many previously published articles, the review can be considered an update of the field and is well written, so its publication is guaranteed.

Author Response

We thank the reviewer for her/his recommendation and agree that the economic uses of halophytes have been reviewed previously. We have endeavoured to make a comprehensive update and address the issue of the effects of salinity on economic use.

Reviewer 2 Report

I have reviewed the article entitled “Economic uses of salt-tolerant plants”

The research study is very interesting and meaningful, this study, the economic and potential economic uses of plants that are listed in the database eHALOPH, including the most tolerant species, halophytes. For nine main categories of economic value, we found a total of 1365 uses amongst all species listed in eHALOPH as of July 2022; this number reduced to 918 amongst halophytes. We did not find any obvious differences in rankings between the more tolerant halophytes and the whole group of salt-tolerant plants, where the order of use was medical, followed by forage, traditional medicine, food and drink, fuel, fuelwood and bioenergy. While many species are potentially important as crops, the effects of salt concentration on their uses are much less well documented. Increasing salt concentration can increase, decrease or have no effect on the concentration of antioxidants found in different species, but there is little evidence on the effect of salinity on potential yield (the product of concentration and biomass). The effect of salinity on forage quality again varies with species, often being reduced, but the overall consequences for livestock production have rarely been evaluated. Salt-tolerant plants have potential uses in the bio-remediation of degraded land (including revegetation, phytoremediation and extraction of NaCl) as well as sources of biofuels, although any use of saline water for the sustainable irrigation of salt tolerant crops must be viewed with extreme caution.

However, the captions in Tables and Figures should be amended.

In addition, English is decent but I suggest a thorough review of the manuscript before accepting it for publication.

To further improve the text, I suggest the following changes in the manuscript.

Abbreviations and acronyms are often defined the first time the term is used within the abstract and again in the main text and then used throughout the remainder of the manuscript. Please consider adhering to this convention. The target journal may have a list of abbreviations that are considered common enough that they do not need to be defined.

My suggestion to ameliorate the manuscript is a general and careful review of the writing, organizing and connecting the main literature cited (with correct references!) in a structured and clear way.

Some general comments which apply to the entire text:

Correctly report scientific plant, crop names

 Correct units

 Correctly report references

Please pay attention on the use of full stop and commas

Abstract;

Modified your abstract; it is only a mere conscript of the study. Better would be to give some introduction followed by the gap in knowledge, hypothesis, general results and then conclusion. The abstract is the only part of the paper that the vast majority of readers see. Therefore, it is critically important for authors to ensure that their enthusiasm or bias does not mislead the reader.

Line 12 database eHALOPH or eHALOPH database?

Line 14; uses amongst all species or uses of all species listed in eHALOPH

Introduction

Introduction section is well presented, not strongly linked to the gaps in the research, therefore the novelty of the work is not significant. Initial lines of introduction are supported by old citations. Please improve the state-of-the-art overview, to clearly show the progress beyond the state of the art. The lack of proper justification creates the wrong impression that the authors are unaware of the recent developments. A high-quality review paper has to provide a proper state-of-the-art analysis after the literature review and only based on the analysis to formulate the paper goals. In addition, the introduction should be clearly stated the research questions and targets first. Then answer several questions: Why is the topic important (or why do you study it)? What are the research questions? What’s the gap of knowledge? Which is the scope of the manuscript? What hypothesis have been made? What has been studied? What are your contributions? The major defect of this study is the debate or argument is not clearly stated in the introduction session. At the end of the introduction, the statement of the paper's goal and the explanation of novelty has to be properly formulated. Currently, this is not performed well. The aim of the introduction should improve

Some mistakes in an introduction. It does not reflect the aim; relevant literature and correlation of this study. The introduction resembles that of a review article and not that of a research article. What’s the gap of knowledge? Which is the scope of the manuscript? What hypothesis have been made? The introduction should be revised accordingly. 

Line 31-32 change to Recently predicted changes in climate and their consequent impact on food production have spawned thousands of publications over the past five years

Line 34; into the core collection of the Web of Science in mid-October 2022)

Line 42; to predict an increase

Line 44; increase by 35-56% by 2050

Line 49; likely to have a negative impact on (change to) adversely affect

Line 59; by analysis change to by an analysis

Results and Discussion.
  Very descriptive Please give only significant results. Also, give mechanistic discussion. It is not a correct way to discuss results based on other scientists' findings. Please elaborate on specified mechanisms which are regulating and result

Conclusion

13.     Add the targeted beneficiary audience who will get benefits from this research.
Also, give clear-cut recommendations

14 In spite this is research article a lack of recent literature (Recent references (last 3 years), therefore the authors should include the most recent references on this subject

15. References

Standardize references

Reviewer 3 Report

Dear authors,
The manuscript Economic uses of salt-tolerant plants is informative and can be helpful for the  readers in planning their any related project. However from my point of view the title is very simple and general. it should be precise and should clearly show reflection of the  manuscript. In addition, the information given is ok but is really limited to medicinal , biofuel and forage. Why not other aspects like food (other than forage)  household , wood and timber and all others which are mentioned in table 3. 
Moreover in every aspect merely the family counting is not sufficient. Authors must have to give  related description about  some genus members of each family as shown in table 6. Though table 6 is also showing only one reference for one family. here number of papers consulted must be more.  The major revision is required to publish the paper. 
